# Ultra-mild bisulfite outperforms existing methods for 5-methylcytosine detection with low input DNA

Qing Dai [1,2,7] ✉, Tanner Baldwin [1,7], Ruitu Lyu [1,7], Bryan Daniels[1,7], Chang Ye [1], Chen Cao[1], Chenyou Zhu[1], Diwen Fan[1], Liane Lin [3], Yushuai Liu[1], Yiding Wang [4] & Chuan He [1,2,5,6] ✉

We present Ultra-Mild Bisulfite Sequencing (UMBS-seq), a method for 5-methylcytosine (5mC) detection that minimizes DNA degradation and background noise. UMBS-seq outperforms conventional bisulfite and enzymatic methyl-sequencing (EM-seq) methods in library yield, complexity, and conversion efficiency when applied to low-input DNA samples. In particular, its effectiveness with low-input cell-free DNA (cfDNA) and hybridization-based target capture highlights its potential for clinical applications, including 5mC biomarker detection and early disease diagnosis.

DNA 5-methylcytosine (5mC) is a key epigenetic mark involved in gene regulation[1,2], and aberrant 5mC patterns are strongly associated with diseases such as cancer[3]. Over the past decades, DNA methylation has emerged as a valuable biomarker for early disease detection, particularly in oncology[4,5]. Mapping 5mC at base resolution is also essential for elucidating the epigenetic mechanisms underlying development, aging, and disease progression. Bisulfite sequencing (BS-seq) is the gold standard for 5mC detection; however, Conventional Bisulfite sequencing (CBS-seq) has suffered from drawbacks, including severe DNA damage, incomplete conversion in regions of high GC content, over-estimation of the 5mC level, and long treatment durations. These drawbacks severely limit its application to low-input or fragmented samples[6], such as cell-free DNA (cfDNA) and Formalin-Fixed Paraffin-Embedded (FFPE) tissue derived DNA[7]. To address these limitations, we recently developed Ultrafast Bisulfite Sequencing (UBS-seq), which substantially shortened reaction times, improved BS efficiency, and reduced false positive rates and over-estimation of the 5mC level[8]. Nevertheless, DNA degradation, particularly problematic for low-input DNA samples, remains a critical challenge.

Two bisulfite-free enzymatic methods for 5mC mapping, TET-assisted pyridine borane sequencing (TAPS) and Enzymatic Methyl sequencing (EM-seq)[8,9], have emerged as non-destructive alternatives to CBS-seq. Among them, EM-seq has shown improved performance over CBS-seq in several key metrics, including higher mapping efficiency, longer insert sizes, lower duplication rates, and reduced GC bias, largely due to its enzymatic conversion strategy[10]. However, EM-seq has several notable limitations: (1) incomplete cytosine conversion, especially when applied to low-input samples[11]; (2) lack of robustness due to enzyme instability; (3) a lengthy and complex workflow; and (4) relatively high reagent costs. In contrast, BS-seq is fast, robust, and automation-compatible. These attributes make it well suited for large-scale clinical applications if the bisulfite-induced DNA damage issue can be resolved. Here we describe Ultra-Mild Bisulfite Sequencing (UMBS-seq), an improved bisulfite conversion method that minimizes DNA damage and background noise while capitalizing on the robustness and efficiency of BS-seq. UMBS-seq outperforms both EM-seq and CBS-seq across key metrics, including library yield/complexity, conversion efficiency, and signal-to-noise ratio, within a streamlined workflow (Fig. 1a). UMBS-seq also causes substantially reduced DNA damage when compared with our previously published UBS-seq method.

[1]Department of Chemistry, The University of Chicago, Chicago, IL, USA. [2]Howard Hughes Medical Institute, The University of Chicago, Chicago, IL, USA. [3]Libertyville High School, Libertyville, IL, USA. [4]Committee on Genetics, Genomics & System Biology, The University of Chicago, Chicago, IL, USA. [5]Department of Biochemistry and Molecular Biology, The University of Chicago, Chicago, IL, USA. [6]Institute for Biophysical Dynamics, The University of Chicago, Chicago, IL, USA. [7]These authors contributed equally: Qing Dai, Tanner Baldwin, Ruitu Lyu, Bryan Daniels. ✉e-mail: daiqing@uchicago.edu; chuanhe@uchicago.edu

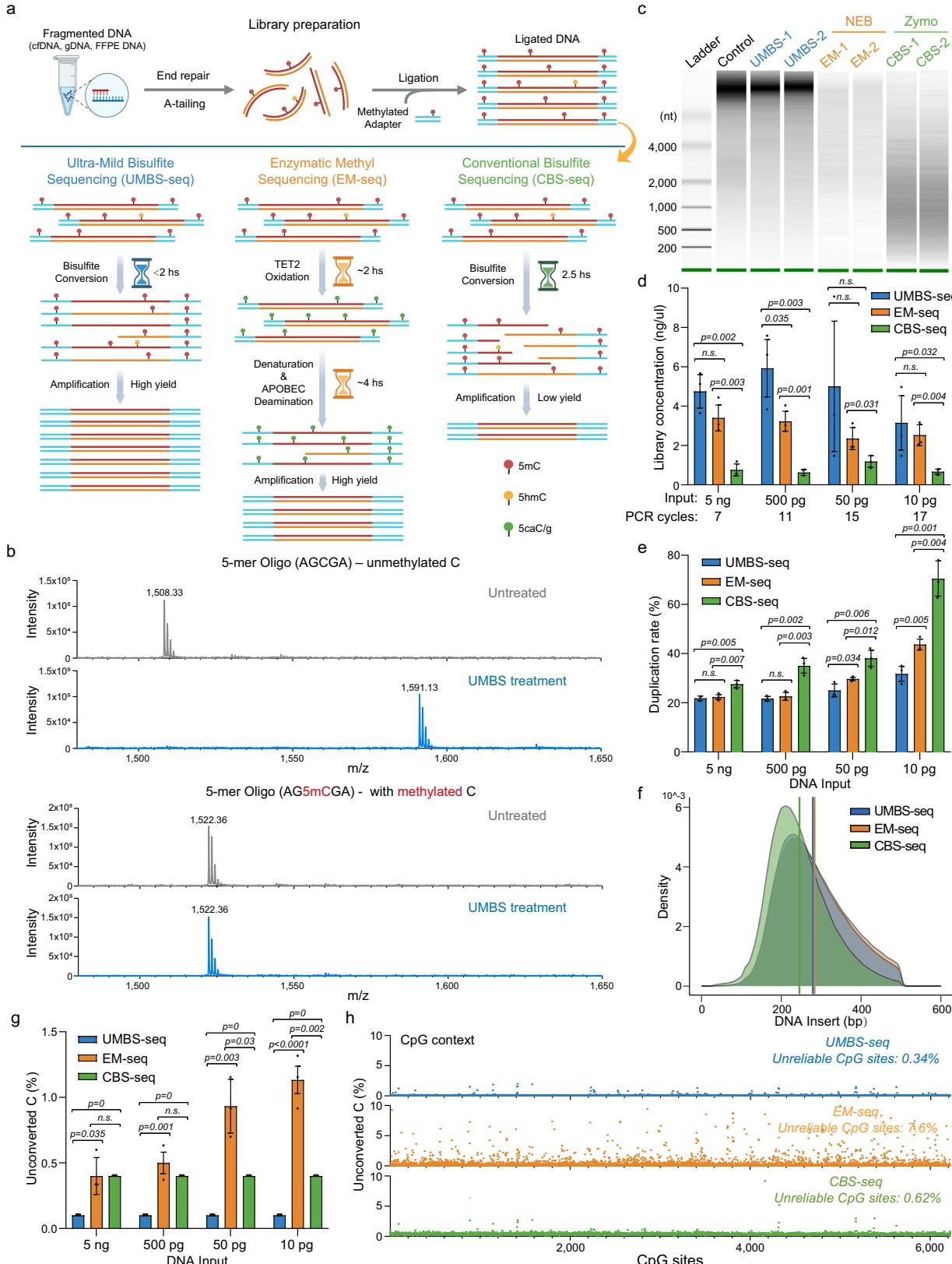

## Results

### Engineering bisulfite reagent composition enables highly efficient C-to-U conversion with reduced DNA damage

Both BS-seq and EM-seq rely on the selective deamination of unmodified C to uracil (U) which is read as T in PCR while 5mC is read as C, enabling base-resolution discrimination between C and 5mC. The efficiency of bisulfite-mediated cytosine deamination is known to be highly dependent on bisulfite concentration, with higher concentrations driving much more efficient conversion[12,13]. Additionally, the reaction pH plays a critical role by facilitating the N3-protonation of cytosines (which is necessary for BS attack on cytosine and the subsequent BS mediated deamination) and by determining the

**Fig. 1 | Ultra-Mild Bisulfite (UMBS) conversion outperforms enzymatic and conventional bisulfite methods across key performance metrics. a** Schematic comparison of the library preparation and conversion workflows for three methyl sequencing methods: Ultra-Mild Bisulfite sequencing (UMBS-seq), Enzymatic Methyl sequencing (EM-seq; NEBNext® Enzymatic Methyl-seq kit, NEB), and Conventional Bisulfite sequencing (CBS-seq; EZ DNA Methylation-Gold kit, Zymo Research). Created in BioRender. Lyu, R. (2025) https://BioRender.com/5m8764l. **b**, MALDI-TOF MS analysis of bisulfite-treated 5-mer DNA oligonucleotides, incubated at 55 °C for 20 minutes, showing a mass shift resulting from the formation of a bisulfite (BS) adduct on cytosine (with subsequent deamination to U-BS), but not on 5mC. **c** Comparative analysis of DNA damage in 50 ng of intact lambda DNA induced by UMBS, EM, and CBS methyl conversion methods, assessed using a bioanalyzer (Agilent RNA 6000 Pico). **d** Concentrations of sequencing libraries generated from the specified lambda DNA amounts after treatment with UMBS, EM, and CBS conversion methods. Library concentration was measured after PCR amplification using the indicated number of cycles with a 15 μL final elution volume. Statistical significance was assessed using two-tailed Student's t-test; p-values are indicated. $N = 3$ libraries per method, prepared from the same DNA input, were analyzed. Data are presented as mean values ± s.d. **e** Duplication rates of sequencing libraries generated from varying lambda DNA amounts, highlighting the superior complexity of UMBS-seq libraries. Statistical significance was assessed using two-tailed Student's t-test; p-values are indicated. Duplication rates of $n = 3$ sequencing libraries per method, each prepared from the same DNA input, were analyzed. Data are presented as mean values ± s.d. **f** DNA insert size distribution of sequencing libraries prepared from 5 ng lambda DNA, demonstrating that UMBS preserves longer DNA insert size with reduced DNA damage. **g** Unconverted C ratios of sequencing libraries generated from varying lambda DNA amounts, showing the high and robust C-to-U conversion efficiency achieved by UMBS-seq. Statistical significance was assessed using two-tailed Student's t-test; p-values are indicated. Unconverted C ratios of $n = 3$ sequencing libraries per method, each prepared from the same DNA input, were analyzed. Data are presented as mean values ± s.d. **h** Scatterplot showing the unconverted C ratios at individual CpG site in 5 ng lambda DNA treated with three different 5mC conversion methods. The average unconverted CpG ratio for each method is indicated.

equilibrium between bisulfite and sulfite species in solution, with bisulfite being the active nucleophile[14]. We hypothesized that maximizing bisulfite concentration at an optimal pH would allow efficient C-to-U conversion under ultra-mild conditions which would minimize DNA damage. To test this, we titrated ammonium bisulfite (72% v/v) with either 20 M KOH or 10 M HCl and identified an optimized formulation consisting of 100 μL of 72% ammonium bisulfite and 1 μL of 20 M KOH. This recipe exhibited a higher conversion efficiency than our previous UBS formulation in a 2-minute test reaction at 55 °C (63.8% vs. 55.6%; Supplementary Fig. 1a). It also achieved complete conversion of a C-containing model DNA oligonucleotide at 55 °C after a 20 min treatment while preserving 5mC integrity (Fig. 1b).

We next screened various reaction temperatures and incubation times using intact lambda DNA[15,16] to evaluate DNA damage by bioanalyzer electrophoresis. The results showed a clear trend with lower reaction temperatures substantially reducing DNA damage despite requiring longer incubation times to reach sufficient converted rates (Supplementary Fig. 1b). Ultimately, we identified an optimal condition at 55 °C for 90 min. An alkaline denaturation step and the inclusion of DNA protection buffer further improves bisulfite efficiency and preserve DNA integrity. We named the modified bisulfite formulation and reaction protocol Ultra-Mild Bisulfite (UMBS) conversion. We leveraged the UMBS recipe and optimized conditions to drastically reduce the DNA damage. When starting from intact lambda DNA, UMBS treatment caused significantly less damage than our recent UBS protocol as demonstrated by fragment size analysis (Supplementary Fig. 2a). In addition, a detailed comparison of libraries prepared with UMBS-seq and UBS-seq showed that UMBS-seq consistently outperformed UBS-seq across all performance metrics except for reaction time, yielding longer insert sizes, higher library yields, greater conversion efficiency, improved GC coverage uniformity, and more accurate DNA methylation estimation (Supplementary Fig. 2b-i).

These data demonstrate the substantial improvements of UMBS over UBS, particularly in reducing DNA damage. To more comprehensively evaluate the performance of UMBS-seq, we compared it to the leading commercially available bisulfite kit (EZ DNA Methylation-Gold Kit, Zymo Research) as well as the leading enzymatic alternative (NEBNext EM-seq kit; New England Biolabs). Using both intact and fragmented lambda DNA as a model, we compared DNA damage induced by UMBS treatment against Enzymatic Methyl-sequencing (EM-seq) and Conventional Bisulfite sequencing (CBS-seq) as benchmarks. UMBS treatment resulted in significantly less DNA fragmentation and higher DNA recovery compared to CBS (Fig. 1c and Supplementary Fig. 1c). Both EM-seq and UMBS-seq largely preserved the integrity of lambda DNA, but EM-seq produced substantially lower DNA recovery, likely due to losses incurred during its multiple purification steps (Fig. 1c and Supplementary Fig. 1c). We next prepared

sequencing libraries using unmethylated lambda DNA ranging from 5 ng to 10 pg and directly compared UMBS-seq, EM-seq, and CBS-seq at each input level. UMBS-seq consistently produced higher library yields across all input levels, indicating improved DNA preservation (Fig. 1d). UMBS-seq libraries displayed substantially higher complexity (lower duplication rates) than CBS-seq libraries and performed better than or comparably to EM-seq (Fig. 1e and Supplementary Data 1). Moreover, the UMBS-seq insert size lengths were comparable to those seen with EM-seq and much longer than those resulting from CBS-seq (Fig. 1f). Additionally, UMBS-seq exhibited significant improvements in CpG coverage uniformity over CBS-seq, and was only slightly worse than EM-seq (Supplementary Fig. 1d).

UMBS-seq consistently generated very low background levels of unconverted cytosines (~0.1%) across all DNA input amounts, with minimal variation even at the lowest inputs. In contrast, CBS-seq showed higher but acceptable background levels (<0.5%), while EM-seq showed significantly higher background signals at lower inputs (exceeding 1% at the lowest input) along with less consistency among triplicates (Fig. 1g). Notably, EM-seq is prone to false positives, with a substantial fraction of unmethylated cytosines (7.6%) exhibiting unconverted ratios greater than 1% (Fig. 1h). We next assessed methylation detection accuracy by analyzing the known methylated CpG sites in pUC19 plasmid DNA. While measured 5mC ratios were consistent (~92%) across all methods at CpG sites, methylation levels at unmethylated CHG and CHH sites appear to be inflated in both CBS-seq and EM-seq, as reflected by their higher unconverted cytosine background (Supplementary Fig. 1e).

The elevated background observed in EM-seq is possibly due to the low concentrations of enzymes employed, which likely limit the rate of enzyme-substrate interactions when substrate concentration is very low. This becomes a particularly noticeable issue at low input levels. In contrast, UMBS-seq uses high concentrations of bisulfite, promoting efficient cytosine conversion even at low inputs. Further analysis revealed that a subset of EM-seq reads displayed widespread failure of C-to-U conversion, with nearly all cytosines remaining unconverted (Supplementary Fig. 3a-b). Introducing an additional denaturation step and filtering out these problematic reads (defined as reads containing more than five unconverted cytosines) both reduced background noise significantly (from 2% to 0.4%), indicating that incomplete DNA denaturation may be another source of false-positive signals in EM-seq (Supplementary Fig. 3c).

## UMBS-seq enables robust and accurate 5mC biomarker detection from low-input cfDNA

To evaluate the clinical applications of UMBS-seq in 5mC biomarker detection, we compared its performance with EM-seq and CBS-seq using cfDNA. UMBS-seq and EM-seq effectively preserved the

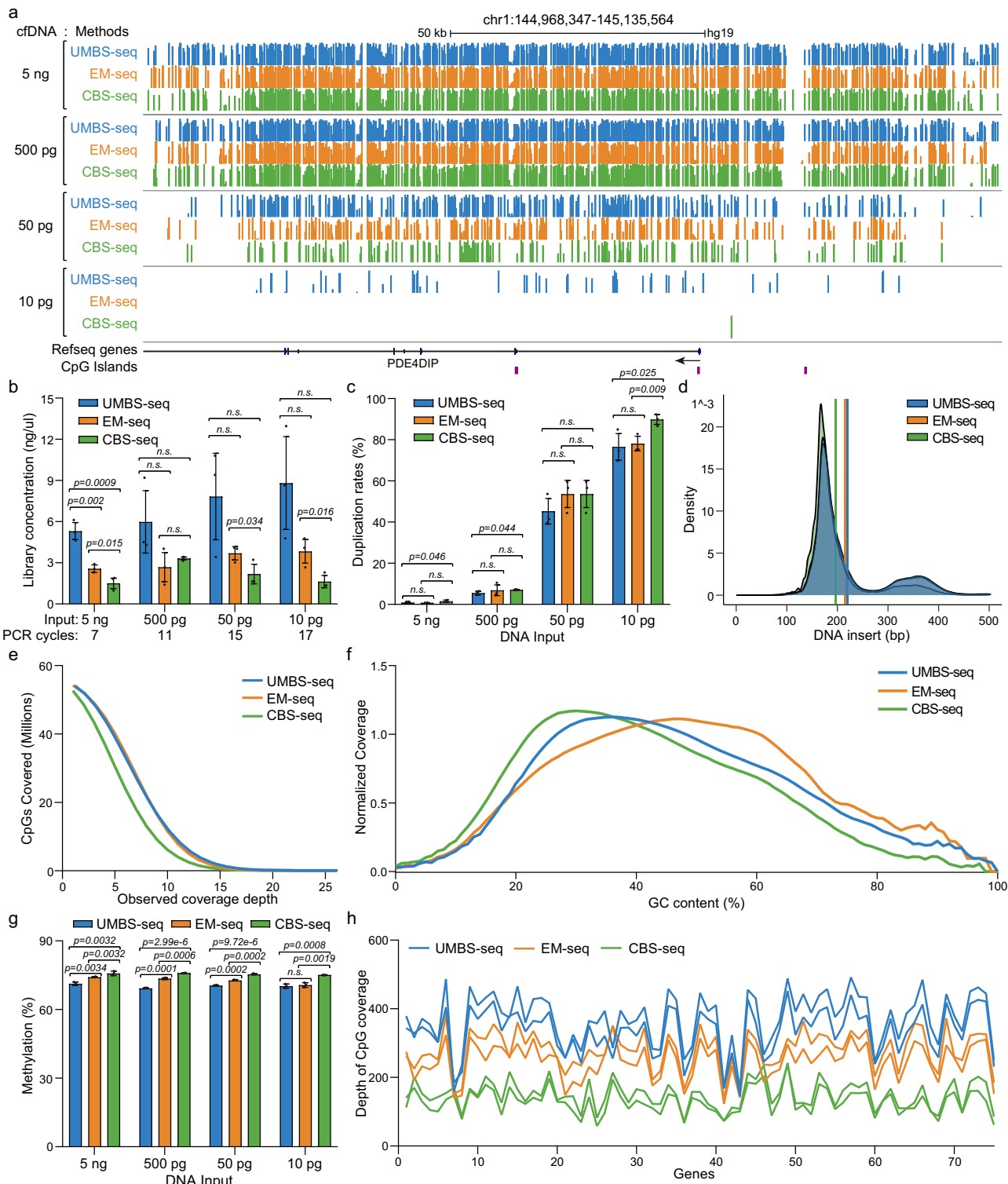

characteristic cfDNA triple-peak profile after treatment when starting from bulk cfDNA whereas UBS-seq did not (Supplementary Fig. 4a). We also built libraries with UMBS, CBS, and EM-seq across a range of low inputs from 5 ng to 10 pg (Fig. 2a). UMBS-seq consistently produced higher library yields and greater complexity than EM-seq at all input levels (Fig. 2b and Supplementary Data 1), along with lower qPCR Ct values (Supplementary Fig. 4b-e) and reduced duplication rates (Fig. 2c). Additionally, bioanalyzer electrophoresis revealed a greater relative abundance of longer cfDNA fragments in UMBS-seq libraries (Fig. 2d), indicating reduced DNA degradation compared to CBS-seq.

Notably, UMBS-seq also performed comparably to EM-seq in terms of its insert size distribution which is typically associated with the non-destructive nature of enzymatic methyl conversion (Supplementary Fig. 4f).

Importantly, both UMBS-seq and EM-seq demonstrated improved genomic coverage and better representation of key genomic features (Supplementary Fig. 4g), particularly in GC-rich regulatory elements such as promoters (Supplementary Fig. 4h) and CpG islands (Supplementary Fig. 4i). Both methods identified more CpG sites than CBS-seq at lower sequencing depths and exhibited superior GC coverage

**Fig. 2 | Comparing UMBS-seq with EM-seq and CBS-seq using low-input cfDNA.** **a** UCSC Genome Browser snapshot showing DNA methylation levels at individual CpG sites across a representative genomic region. cfDNA libraries were prepared with UMBS-seq, EM-seq, and CBS-seq methods using varying cfDNA input amounts. **b** Concentration of sequencing libraries generated from varying cfDNA amounts treated with the three different conversion methods. Concentrations were measured after the indicated number of PCR cycles with a 15 μL elution volume. Statistical significance was assessed using two-tailed Student's *t*-test; *p*-values are indicated. *N* = 3 libraries per method, prepared from the same DNA input, were analyzed. Data are presented as mean values ± s.d. **c** Duplication rates of sequencing libraries generated from varying input amounts of cfDNA, highlighting the superior library complexity achieved with UMBS-seq. Statistical significance was assessed using two-tailed Student's t test; *p*-values are indicated. Duplication rates of *n* = 3 sequencing libraries per method, each prepared from the same DNA input, were analyzed. Data are presented as mean values ± s.d. **d** DNA insert size

distribution of 10 pg cfDNA libraries, showing that UMBS-seq and EM-seq preserve larger insert sizes and better maintain cfDNA integrity compared to CBS-seq. **e** Number of identified CpG sites in cfDNA libraries prepared from 5 ng of input, showing that UMBS-seq detects more CpG sites than CBS-seq at higher sequencing coverage depth. **f** GC-bias plot for UMBS-seq, EM-seq and CBS-seq cfDNA libraries prepared from 5 ng of input. UMBS-seq and EM-seq data display better GC coverage uniformity compared to CBS-seq data. **g** Bar plot showing the 5mC levels in CpG contexts of cfDNA compared across libraries prepared using UMBS-seq, EM-seq, or CBS-seq. Statistical significance was assessed using two-tailed Student's *t*-test; *p*-values are indicated. CpG methylation levels of *n* = 3 sequencing libraries per method, each prepared from the same DNA input, were analyzed. Data are presented as mean values ± s.d. **h** The combination of the Nanodigmbio μCaler capture kit with UMBS-seq on 5 ng cfDNA from healthy individuals results in enhanced CpG coverage compared to its combination with EM-seq and CBS-seq.

uniformity (Fig. 2e,f), enabling more comprehensive CpG methylation profiling with fewer spurious methylation calls. Consistent with our findings from lambda DNA, UMBS-seq achieved lower background methylation levels at sparsely methylated CHG and CHH sites compared to EM-seq and CBS-seq across all tested input amounts (Supplementary Fig. 5a-c). At CpG sites, the methylation level of EM-seq and CBS-seq is inflated relative to UMBS-seq, suggesting that they overestimate the methylation level due to their incomplete conversion (Fig. 2g). Notably, each method exhibited high reproducibility both across technical replicates and between distinct conversion approaches (Supplementary Fig. 5d-h), and accurately reconstructed known DNA methylation patterns within promoter regions and gene bodies starting with 5 ng cfDNA (Supplementary Fig. 5i-k).

To further assess the potential of UMBS-seq in disease biomarker discovery, we applied the μCaler Total Solution for Methylation capture technology (μCaler EMS Panel v1.0, Nanodigmbio) to enrich cfDNA fragments containing CpG sites associated with multiple cancer-related genes. Successful targeted methylation capture with this method relies on both efficient C-to-U conversion and preservation of cfDNA fragment integrity to ensure high specificity and efficiency of capture probes[17]. When coupled with UMBS-seq, the μCaler capture achieved significantly greater coverage of targeted CpG sites compared to EM-seq and CBS-seq (Fig. 2h and Supplementary Data 2), underscoring the suitability and advantage of UMBS-seq for targeted methylation profiling in clinical biomarker applications, particularly when working with challenging low-input cfDNA samples.

Collectively, our results demonstrate that UMBS-seq outperforms EM-seq across multiple key performance metrics, including library yield, complexity, C-to-U conversion efficiency, and, notably, workflow simplicity and robustness. UMBS-seq also consistently exhibits markedly improved performance over CBS-seq across all evaluated parameters. A summary of the relative strengths and weaknesses of the four tested methods, UMBS-seq, UBS-seq, EM-seq, and CBS-seq can be found in Supplementary Fig. 6. While UBS can replace most routine bisulfite reactions, we anticipate that UMBS-seq will enable broader clinical adoption of BS-seq by facilitating robust, sensitive, and accurate detection of 5mC levels in low-input cfDNA samples, setting a standard for 5mC biomarker discovery and detection.

## Discussion

UMBS-seq represents a significant advance in bisulfite conversion by minimizing DNA damage while maintaining efficient cytosine conversion. Compared with CBS-seq, it consistently delivers higher library complexity, improved CpG coverage, and more accurate methylation measurements. Unlike EM-seq, UMBS-seq avoids enzyme-related variability, elevated background noise at low inputs, and workflow complexity. When combined with targeted capture, UMBS-seq further enables sensitive and specific detection of clinically relevant 5mC biomarkers from low-input cfDNA. Together, these features establish

UMBS-seq as a robust and scalable platform that combines the accuracy of bisulfite chemistry with the practicality needed for translational and clinical applications.

## Methods
### Maldi
The Matrix Assisted Laser Desorption Ionization Time of Flight Mass Spectrometry (MALDI TOF MS) assay was performed treating synthetic oligonucleotides (having sequences 5′-AGCGA-3′ or 5′-AG5mCGA-3′) with an UMBS variation consisting of 100 μL 72% ammonium bisulfite and 1 μL 20 M KOH. Exactly 9 μL of the preprepared recipe was mixed with 1 μL of the DNA oligo. *N* = 1 oligos sample was treated at 55 °C for up to 20 min. Afterwards, 1.6 μL of the BS/oligo reaction mixture was combined with 1.6 μL of a matrix consisting of 2,4,6-Trihydroxyacetophenone monohydrate on a MALDI plate. The MALDI-TOF MS spectrum was recorded on a Bruker Ultra-flex TOF/TOF MALDI mass spectrometer using the negative ion reflection mode. Data was processed in the Bruker Flex Analysis software 3.4.

### Key condition differences between UMBS-seq and UBS-seq workflows
There are four key differences between UMBS-seq and our previously published UBS-seq. First, the UMBS-seq recipe includes a DNA protection buffer containing an organic solvent and antioxidant to prevent DNA renaturation and reduce DNA damage. This component is absent in UBS-seq. Second, the method of pH titration differs. UBS-seq achieves a desired pH by mixing two commercially available ammonium bisulfite solutions (70% and 50%), each with distinct pH values. UMBS-seq uses only 70% ammonium bisulfite titrated with a small volume of 20 M KOH. This approach enables milder bisulfite conversion at lower temperatures by slightly increasing the bisulfite concentration and provides greater flexibility for adjusting reaction conditions according to each user's needs. Third, UMBS-seq employs a "lower temperature, longer reaction time" strategy (55 °C for 90 min instead of 98 °C for 10 min), which minimizes DNA damage while maintaining high conversion efficiency. Finally, DNA denaturation in UBS-seq occurs at the high bisulfite reaction temperature, whereas in UMBS-seq, DNA is pre-denatured by a 20 min incubation at 42 °C in 0.3 M KOH, a gentler method that avoids the damaging effects of high-temperature treatment.

### Lambda DNA library preparation
Two samples of ~250 ng of unmethylated Lambda genomic DNA (-dcm, -dam) from Thermo Fisher were fragmented with NEBNext UltraShear for 25–30 min according to the NEB protocol. The samples were then pooled together and purified with Cytiva magnetic beads with a 1.0x beads ratio following the recommended protocol. The remaining material was taken for subsequent end repair, A-tailing, and ligation of a methylated sequencing adapter with the KAPA EvoPrep Kit. The NEB

single molecule Methylated Adapter was used during ligation, and 3 µL of USER enzyme was added to the ligation reaction mixture following ligation and heated at 37 °C for 15 min to open the NEB Methylated Adapter loop. A final magnetic beads (Cytiva Lifesciences) purification with a 0.8x beads ratio was used to purify the adaptor ligated dsDNA. After quantification with a bioanalyzer, the concentration of the prepared material was adjusted and a series dilution was made resulting in samples of 2 µL each with total amounts of material at 5 ng, 0.5 ng, 50 pg, and 10 pg. The 2 µL samples at each input were then adjusted to the required volumes to begin EM-seq or Zymo Gold treatments according to their respective protocols, or they began UMBS-seq treatment.

EM-seq (NEBNext® Enzymatic Methyl-seq kit, NEB) and CBS-seq (EZ DNA Methylation-Gold kit, Zymo Research) experiments were performed following the manufacturer's protocols except for a modified denaturing process to mitigate the issue of low unconversion ratios due to inadequate denatured DNA. After TET oxidation, 20 M KOH was added to the reaction mixture to a final concentration of 0.5 M and heated at 42 °C for 20 min before being placed on ice. The EM-seq samples were purified with an Oligo Clean and Concentrator kit (Zymo) before proceeding with a second denaturation by formamide and deamination by APOBEC according to the NEB EM-seq protocol. CBS-seq samples were treated according to Zymo EZ DNA methylation-Gold™ Kit manual. For UMBS-seq treatment, 3 µL of 0.5 M KOH was added to the 2 µL of sample containing DNA (final conc. 0.3 M KOH). The samples were denatured under these alkaline conditions at 42 °C for 20 minutes before being placed immediately on ice. In separate PCR tubes, the UMBS recipe was made by slowly adding 1 µL of 20 M KOH to 36.5 µL of 72% ammonium bisulfite (Ellis Bio), briefly vortexing to mix, then adding 7.5 µL of DNA Protection Buffer (Qiagen), and vortexing to mix one final time. The previously prepared 5 µL of denatured DNA was then added to the 45 µL UMBS recipe mixture and heated at 55 °C for 90 minutes before being removed and left at room temperature (not placed on ice). The 50 µL UMBS reaction mixture was transferred into 600 µL of M-Binding buffer (Zymo Research) waiting in fresh 1.5 mL Eppendorf tubes and mixed by pipetting. The 650 µL mixture was then transferred to Zymo Spin Columns and spun at maximum speed for 30 s (all centrifugation steps were carried out at maximum speed for 30 s and the collection tube was emptied only as needed). The DNA bound column was washed once with 100 µL Zymo M-Washing buffer (Zymo Research), then 200 µL of M-Desulphonation buffer (Zymo Research) was added and allowed to sit on the column for 15 minutes at room temperature. After centrifugation, a total of four washing steps were completed using 200 µL of Zymo M-Wash buffer each time. A final 2-minute centrifugation of the empty, DNA bound columns at maximum speed was used to fully dry the columns before elution with 23 µL of distilled water.

Both EM-seq and CBS-seq treated samples were also eluted with 23 µL of distilled water at the termination of their respective treatment protocols. Real-time qPCR was then performed by mixing 2 µL of each sample with 5 µL distilled water, 1 µL 20x SYBR Green I dye from Thermo Fisher, 1 µL of NEB Universal primer, 1 µL of NEB Index primer, and 10 µL of KAPA HiFi U+ Master Mix. The qPCR was run on a Roche LightCycler 96. Subsequent PCR was completed by combining the remaining 20 µL of converted DNA with 25 µL of KAPA HiFi U+ Master Mix and 5 µL of NEB Dual Index Primers.

## Lambda DNA damage assay
Approximately 50 ng of intact lambda DNA (-dcm, -dam) was used as input for each sample. The DNA was subjected to methylation conversion using the UMBS, EM, and CBS methods, following previously described protocols. The resulting products were analyzed using a bioanalyzer (Agilent RNA 6000 Pico) to assess DNA integrity. As lambda DNA was denatured into ssDNA during the conversion process,

a single-stranded nucleotide-binding dye was employed to visualize the distribution and size range of the resulting fragments.

## cfDNA library preparation
Cell-free DNA (cfDNA) was extracted and purified from pooled human plasma (Innovative Research Inc., Cat. #IPLAWBK2E) using the Qiagen QIAamp Circulating Nucleic Acid kit followed by Ampure beads size selection to remove gDNA contamination. End repair, A-tailing, and ligation of a sequencing adapter were completed with the KAPA Evo-Prep Kit (Roche) and the NEB Methylated Adaptor in an identical manner to that described for the Lambda DNA library preparation. The Bioanalyzer electrophoresis confirmed the expected triple peak profile of cfDNA. cfDNA samples were diluted and prepared for treatments in an identical manner to that described above for lambda DNA. Conversion treatments were also carried out in the same manner as described for the lambda DNA.

## µCaler targeted methylation capture
The µCaler EMS Panel v1.0 (Early Methylation for Screening) covers methylation gene sites (nearly 2,500 CpG sites) related to nine major high incidence cancers, including selected sites approved by the NMPA and FDA. cfDNA was extracted from plasma using a magnetic bead-based protocol optimized for low-input samples. After adaptor ligation, Methyl conversion was performed using one of the following methods: UMBS-seq, EM-seq, or CBS-seq, with 5 ng of cfDNA input. Following conversion, DNA was purified and amplified by PCR with different index. Then, subjected to targeted capture using the µCaler EMS Panel v1.0 according to the manufacturer's instructions. Final libraries were quantified using a bioanalyzer, pooled, and sequenced on an Illumina platform with paired-end 150 bp reads. Raw sequencing reads were trimmed to remove adapters and low-quality bases using Trim Galore (v0.6.10), then aligned to the human reference genome (hg38) using Bismark (v0.24.0). Coverage at each CpG site within the panel was calculated and normalized based on the number of uniquely mapped reads. For visualization and quantification, the average coverage was computed across all CpG sites within each targeted gene in the µCaler EMS Panel v1.0.

## Processing and quality control of sequencing data
Raw paired-end bisulfite sequencing reads of three methyl conversion methods were trimmed using Trim Galore (v 0.6.10)[18], which removes Illumina adapter sequences and low-quality bases (Phred score <20). The trimmed reads were then aligned to the reference genome using bismark (v0.24.0)[19]. Prior to alignment, the reference genomes (e.g., pUC19 DNA, lambda DNA, and hg19) were prepared using bismark_genome_preparation, which generates bisulfite-specific indices. Mapping of trimmed sequencing data was performed in paired-end mode with default parameters, and alignment outputs were stored in BAM format for downstream processing. Following alignment, PCR duplicates were removed using deduplicate_bismark to minimize amplification bias. To eliminate unconverted clusters in the EM-seq data, reads with more than 5 unconverted C sites left unconverted were discarded using custom Python scripts. Methylation levels were extracted from the deduplicated BAM files using bismark_methylation_extractor with the --cytosine_report, --CX, and --bedGraph options enabled. This step generated methylation calls for cytosines in CpG, CHG, and CHH contexts, as well as genome-wide coverage files and per-base resolution methylation reports. The methylation extraction was performed using a buffer size of 10 G to ensure efficient processing of large files. All downstream data analysis, visualization, and quality control were conducted using custom Python and R scripts.

## GC coverage analysis
GC coverage analysis was performed using Picard Tools (v2.18.14)[20]. The CollectGcBiasMetrics module was used to assess the distribution

of sequencing coverage across the genome as a function of GC content (picard CollectGcBiasMetrics I=deduplicated.sort.bam O=gc_bias_metrics.txt CHART=gc_bias_metrics.pdf S=summary_metrics.txt R=hg19.fa). Input BAM files from aligned and deduplicated bisulfite reads were analyzed to calculate normalized coverage across genomic regions with GC content ranging from 0% to 100%. The resulting output files included summary metrics and GC bias plots, which were used to evaluate GC coverage uniformity.

## Visualization of DNA methylation on cytosines

To visualize CpG methylation patterns across the genome, bedGraph files containing DNA methylation data (using the same sequencing depth) for CpG sites with a minimum read coverage of 10 were uploaded to the UCSC Genome Browser. deepTools (v3.5.1) was used to generate metagene profile plots for both DNA methylation and reads coverage. For gene-level visualization, methylation levels were scaled across gene bodies to normalize for gene length. Additionally, average read coverage was plotted around CpG islands with ±1 kb flanking "shore" regions, and promoter regions, defined as 3 kb upstream and 3 kb downstream of transcription start sites (TSS).

## Statistics and reproducibility

The specific statistical tests used for data analysis are detailed in the corresponding figure legends. To compare methylation ratios among triplicates generated from the three methylation conversion methods, CpG methylation levels were calculated within 10 kb sliding windows across the genome. All UMBS-seq, EM-seq, and CBS-seq experiments in this study were performed with at least three independent technical replicates, unless otherwise noted. No statistical method was used to predetermine sample size.

## Data availability

The raw and processed data of UMBS-seq, EM-seq, and CBS-seq experiments performed using lambda DNA, cell-free DNA (cfDNA), and methylated pUC19 DNA have been deposited in the Gene Expression Omnibus under the accession number: GSE294197. Source data are provided with this paper.

## Code availability

All the bioinformatic scripts used in this study are available at https://github.com/Ruitulyu/UMBS-seq under the MIT License, an open-source license approved by the Open Source Initiative[21].

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

## Acknowledgements

We thank the functional genomics facility at the University of Chicago for performing high-throughput sequencing. This work was supported by the US National Institutes of Health (HG006827 to C.H.). C.H. is an investigator of the Howard Hughes Medical Institute.

## Author contributions

Q.D. and C.H. conceived the project. Q.D. and T.B. developed the UMBS experimental procedures. T.B. performed all UMBS, EM-seq, and CBS-seq experiments with assistance from C.C., C.Z., D.F., L.L., Y.L., and Y.W. Bioinformatic analysis of all high throughput sequencing data was completed by R.L. and B.D. with the help of C.Y. The manuscript was written by Q.D., T.B., R.L., and C.H. with input from all authors.

## Competing interests

A patent application on UMBS-seq has been filed by the University of Chicago Polsky Center for Entrepreneurship and Innovation. C.H. is a scientific founder, a member of the scientific advisory board and equity holder of Aferna Bio, AllyRNA, and Ellis Bio, a scientific cofounder and equity holder of Accent Therapeutics, and a member of the scientific advisory board of Rona Therapeutics and Element Biosciences. Q.D. holds equity in Ellis Bio Inc. R.L. serves as scientific advisor to Ellis Bio Inc. The remaining authors declare no competing interests.
