## [Transparent Peer Review file · Nature Communications]

Ultra-mild bisulfite outperforms existing methods for 5-methylcytosine detection with low input DNA

Corresponding Author: Professor Chuan He

Version 0:

Reviewer comments:

Reviewer #1

(Remarks to the Author)

Comments: The authors amended the Conventional Bisulfite Sequencing (CBS-seq) method and developed an Ultra-Mild Bisulfite Sequencing (UMBS-seq) method for the genome wide mapping of 5mC.

Specific comments:

1. The authors reported an UBS-seq method in 2024, which was also an amended method of CBS-seq. The UBS-seq also does not cause DNA damage and could be used in the mapping of 5mC in cfDNA. Since that UBS-seq and UMBS-seq are both amended methods of CBS-seq, systematic comparisons between these two methods, including DNA degradation, library yield, and cytosine conversion efficiency, are required to demonstrate the importance of developing UMBS-seq, a method with slight modifications compared to UBS-seq.
2. A table of the comparison between UMBS-seq and other methods should be provided to make it easier for the readers to understand the superiority of UMBS-seq.
3. UBS-seq can realize the effective deamination of cytosine within 3 min, greatly diminishing the time of DNA exposed to bisulfite, which was the reason of reducing DNA degradation in bisulfite treatment. UMBS-seq required a much longer time (2 hours) for bisulfite treatment, which might cause more DNA degradation than UBS-seq.
3. A detail description of the different conditions used in UBS-seq and UMBS-seq should be provided to highlight the improvements in UMBS-seq.

Reviewer #2

(Remarks to the Author)

This study presents an Ultra-Mild Bisulfite Sequencing (UMBS-seq) method that effectively improves conversion efficiency while reducing DNA damage and background noise. The method demonstrates strong performance on ultra-low input samples and cfDNA, indicating significant potential for clinical diagnostics. However, the author's team published a similar paper on "UBS-seq" in 2024, and the current manuscript lacks a direct comparison with this previous work. The following points should be addressed:

1. Figure 1c: The authors compare the degree of DNA damage across the different methods. However, in the provided gel image, the band intensity for the EM-seq lane appears faint, which seems inconsistent with the expected amount of DNA. Could the authors provide the corresponding electropherogram (e.g., a peak graph from a Bioanalyzer) to further substantiate this result?
2. Figure 1d: While UMBS-seq generates a higher library yield with higher DNA input, its performance at lower inputs of 50 pg and 10 pg shows significant variability and, in some cases, appears to be less effective than EM-seq. Could the authors provide an explanation for this observation?
3. The study uses lambda DNA as the standard sample to demonstrate the reduced damage and optimized conditions of UMBS-seq. However, two issues arise from this:
(1) The uniform length of lambda DNA may not fully represent the behavior of more heterogeneous samples like cfDNA. The authors are encouraged to provide a comparative analysis showing the effects of the different methods on cfDNA itself to

demonstrate the preservation of DNA integrity.

(2) In a typical library preparation workflow, DNA is first fragmented and then ligated with adapters. These fragments, much like cfDNA, are already in a small size range. When these short fragments undergo bisulfite conversion, is the resulting DNA damage as severe as that observed with the initially intact, much larger lambda DNA? A clarification on this point would be helpful.

4. I noted that Prof. He's lab published a paper on UBS-seq in 2024 (DOI: 10.1038/s41587-023-02034-w), which introduced a solution for ultra-fast bisulfite conversion. Could the authors elaborate on the similarities and differences between UBS-seq and the current UMBS-seq? A detailed comparison of their experimental protocols, primary application areas, and respective limitations would be highly valuable.

Version 1:

Reviewer comments:

Reviewer #1

(Remarks to the Author)

The authors addressed my concerns.

Reviewer #2

(Remarks to the Author)

After reviewing the manuscript, I am pleased to suggest that I have no revisions or objections to raise. The content is well-presented, and it contributes valuable insights to the field. Therefore, I recommend proceeding with its acceptance.

Point-by-Point Response

Summary of Revision

We thank all reviewers for their thoughtful and constructive comments on our manuscript, which were very useful in revising and strengthening it. All changes in the manuscript are marked in blue. The following is a summary of key experiments and analyses we have performed during revision based on the reviewers' comments:

1. We conducted a systematic side-by-side comparison of UMBS-seq and UBS-seq by preparing libraries from fragmented mESC gDNA, supplemented with unmethylated lambda DNA and methylated pUC19 DNA as spike-ins. Our results demonstrate that UMBS-seq offers clear advantages over UBS-seq, including markedly reduced DNA damage, higher library yield, more accurate DNA methylation estimation, improved GC coverage uniformity, and consistently high conversion efficiency.
2. We performed comparative analyses using cfDNA treated with UMBS, EM, and UBS conversion methods. Fragment size profiles showed that UMBS and EM preserved the characteristic cfDNA size distribution, whereas UBS caused substantially severer degradation. These findings confirm that UMBS-seq preserves cfDNA integrity at a level comparable to that observed with lambda DNA.
3. We carried out a side-by-side comparison of UMBS, EM, and CBS conversion methods using fragmented lambda DNA analyzed with the Agilent Bioanalyzer. We found DNA damage was less pronounced overall in these shorter fragments than in larger intact lambda DNA due to their reduced length and fewer potential cleavage sites. Nevertheless, UMBS and EM caused significantly less DNA damage than the CBS method.
4. We summarized four key methodological differences between UMBS-seq and UBS-seq, including DNA denaturation process, DNA protection buffer, temperature and reaction time, and pH titration method, and added a new paragraph to the Methods section to help readers more clearly understand these distinctions.

Reviewer #1:

Comments for the Author:

The authors amended the Conventional Bisulfite Sequencing (CBS-seq) method and developed an Ultra-Mild Bisulfite Sequencing (UMBS-seq) method for the genome wide mapping of 5mC.

Specific comments:

1. The authors reported a UBS-seq method in 2024, which was also an amended method of CBS-seq. The UBS-seq also does not cause DNA damage and could be used in the mapping of 5mC in cfDNA. Since UBS-seq and UMBS-seq are both amended methods of CBS-seq, systematic comparisons between these two methods, including DNA degradation, library yield, and cytosine conversion efficiency, are required to demonstrate the importance of developing UMBS-seq, a method with slight modifications compared to UBS-seq.

Response: We thank the reviewer for the helpful suggestion. To systematically compare UBS-seq and UMBS-seq, we conducted a side-by-side evaluation by preparing libraries from fragmented mESC gDNA containing unmethylated lambda DNA and methylated pUC19 DNA spike-ins. All libraries were constructed from the same amount of DNA and subjected to 9 cycles of PCR amplification. We then assessed the performance metrics mentioned by the reviewer. Our findings demonstrate that:

1. UMBS-seq caused markedly less DNA damage than UBS-seq, as evidenced by the Agilent Tapstation assay (Response Fig. 1a), and UMBS-seq prepared libraries exhibited larger insert sizes when using fragmented mESC gDNA compared to UBS-seq (Response Fig. 1b).
2. UMBS-seq libraries achieved over twice the library concentration of UBS-seq libraries (Response Fig. 1c).
3. Both UMBS-seq and UBS-seq exhibited >99.5% conversion rates (UMBS-seq: 99.8%, UBS-seq: 99.7%) with very low background, defined as cytosine sites with unconverted C-T ratios greater than 1% (Response Fig. 1d-e).
4. UMBS-seq provided higher accuracy in methylation detection, with fewer false negatives based on the methylated CpG sites in pUC19 DNA (Response Fig. 1f-g).
5. UMBS-seq reads demonstrated a more uniform distribution and lower GC Dropout across GC content (Response Fig. 1h-i).

These results highlight the advantages of the UMBS-seq method over UBS-seq across various performance metrics, including significantly reduced DNA damage, enhanced library yield, more accurate DNA methylation estimation, improved GC coverage uniformity, and higher conversion efficiency. We have also included these data in Extended Data Figure 2, with a detailed description highlighted in blue in the main manuscript.

Response Fig. 1: **a**, Comparative analysis of DNA damage in 100 ng of intact lambda DNA induced by untreated, UMBS, UBS methyl conversion methods, assessed using a bioanalyzer (Agilent RNA 6000 Pico). **b**, DNA insert size distribution of sequencing libraries prepared from 10 ng mESC gDNA, demonstrating that UMBS-seq preserves longer DNA insert sizes than UBS-seq due to its reduced DNA damage. **c**, Concentrations of sequencing libraries generated from the specified lambda DNA amounts after treatment with either UMBS or UBS conversion. Library concentration was measured after 9 PCR cycles with a 20 μ L final elution volume. Statistical significance was assessed using Student's t-test ($n = 2$); p-values are indicated. **d**, Scatterplot showing the unconverted C ratios at individual CpG sites in lambda DNA treated with either UMBS or UBS conversion. **e**, Bar plot showing the average unconverted C-to-T ratios of cytosines in both CpG and non-CpG contexts in lambda DNA following either UMBS-seq or UBS-seq. **f**, Cleveland dot plot showing the estimated DNA methylation levels at each CpG site of methylated pUC19 DNA

following either UMBS-seq or UBS-seq. **g**, Bar plot showing the average DNA methylation levels across all CpG sites in methylated pUC19 DNA following either UMBS-seq or UBS-seq. **h**, GC-bias plot for UMBS-seq and UBS-seq libraries prepared from 10 ng of mESC gDNA. UMBS-seq shows better GC coverage uniformity compared to UBS-seq. **i**, Bar plot showing the GC dropout ratios in UMBS-seq and UBS-seq libraries prepared using mESC gDNA.

2. A table of the comparison between UMBS-seq and other methods should be provided to make it easier for the readers to understand the superiority of UMBS-seq.

Response: We appreciate the reviewer’s suggestion. In response, we have included the following table, which provides a detailed comparison of the performance of UMBS-seq against UBS-seq, EM-seq, and CBS-seq across nearly all performance metrics, including robustness, input DNA range, protocol time, DNA damage, false positives, and library complexity. We have also included this table as Extended Data Fig. 6 in the main manuscript to further highlight the superiority of UMBS-seq over the other tested methods.

	UMBS-seq	UBS-seq	EM-seq	CBS-seq
Robustness	High	High	Low	High
Input DNA	>0.01 ng	>1 ng	>0.1 ng	>50 ng
Protocol time	2-3 h	35 minutes	7-8 h	3-4 h
DNA damage	Low	Medium	Low	High
False positives	Near zero	Near zero	High	Moderate
Library Complexity	High	Medium	High	Low

Response Figure 2: Comparison of UMBS-seq with UBS-seq, EM-seq, and CBS-seq based on various performance metrics, including robustness, input DNA range, protocol time, DNA damage, false positives, and library complexity.

3. UBS-seq can realize the effective deamination of cytosine within 3 min, greatly diminishing the time of DNA exposed to bisulfite, which was the reason of reducing DNA degradation in bisulfite treatment. UMBS-seq required a much longer time (2 hours) for bisulfite treatment, which might cause more DNA degradation than UBS-seq.

Response: We appreciate the reviewer’s insightful comment. In UBS-seq, bisulfite (BS) treatment is performed at 98 °C to accelerate the reaction, enabling BS conversion to be completed within a very short time and thereby reducing DNA degradation. Additionally, the high temperature helps denature the double-stranded DNA (dsDNA), facilitating BS conversion. However, a very high temperature, even for a short treatment time, can still induce significant DNA damage. While a quick UBS treatment at 98 °C reduces DNA damage compared to conventional bisulfite (CBS) treatment to some extent, DNA damage remains a concern, particularly for low-input clinical samples.

In UMBS-seq, we have recently discovered a new bisulfite recipe that demonstrates

even higher BS conversion efficiency than UBS-seq (Extended Data Fig. 1a). Further optimization of reaction conditions revealed that DNA damage can be minimized at lower temperatures despite the longer treatment time needed to ensure high conversion efficiency (Extended Data Fig. 1b). To ensure complete denaturation of dsDNA at lower temperatures and prevent renaturation, we incorporated an additional alkaline denaturation step before bisulfite conversion and included a DNA-protecting buffer in the bisulfite treatment reaction mixture. This approach allows for very high BS conversion efficiency with significantly reduced DNA damage when treated at 55 °C for 90 minutes. Our optimized conditions showed substantially less DNA damage after UMBS-seq compared to UBS-seq, as demonstrated in Response Figure 1a-c. We recommend that UBS is best used to replace all common lab bisulfite reactions, whereas UMBS is best suited for low-input clinical samples such as cfDNA. We added this to the last paragraph in the discussion.

4. A detailed description of the different conditions used in UBS-seq and UMBS-seq should be provided to highlight the improvements in UMBS-seq.

Response: We thank the reviewer for this helpful suggestion. In summary, there are four key differences between the UBS-seq¹ and UMBS-seq conditions that highlight the improvements made in UMBS-seq:

1. DNA protection buffer: The UMBS-seq recipe includes a DNA protection buffer composed of an organic solvent and antioxidant to prevent DNA renaturation and reduce DNA damage, whereas the UBS-seq recipe lacks these components.
2. pH titration method: UBS-seq achieves the optimal pH by mixing two commercially available ammonium bisulfite solutions (70% and 50%) with different pH values. In contrast, UMBS-seq uses 70% ammonium bisulfite titrated with a small volume of 20 M KOH. This method allows mild bisulfite conversion at much lower temperatures and offers greater flexibility for adapting reaction conditions.
3. Temperature and reaction time: UBS-seq is performed at 98 °C for 10 min, whereas UMBS-seq is conducted at 55 °C for 90 min. This “lower temperature, long reaction time” condition minimizes DNA damage while maintaining high conversion efficiency.
4. DNA denaturation process: In UBS-seq, DNA denaturation occurs automatically due to the high reaction temperature (98 °C). In UMBS-seq, DNA is pre-denatured by a 20 min incubation at 42 °C in 0.3 M KOH, a method that causes minimal damage and avoids the harsher high-temperature treatment.

We have incorporated these detailed differences between the UMBS-seq and UBS-seq recipes and workflows into the Methods section, under the paragraph titled “*Key condition differences between UMBS-seq and UBS-seq workflows*” for clarity, and added specific application recommendations to the last paragraph.

Reviewer #2:

Comments for the Author:

This study presents an Ultra-Mild Bisulfite Sequencing (UMBS-seq) method that effectively improves conversion efficiency while reducing DNA damage and background noise. The method demonstrates strong performance on ultra-low input samples and cfDNA, indicating significant potential for clinical diagnostics. However, the author's team published a similar paper on "UBS-seq" in 2024, and the current manuscript lacks a direct comparison with this previous work.

Response: We thank the reviewer for the positive feedback regarding the clinical potential of UMBS-seq.

Specific comments:

1. Figure 1c: The authors compare the degree of DNA damage across the different methods. However, in the provided gel image, the band intensity for the EM-seq lane appears faint, which seems inconsistent with the expected amount of DNA. Could the authors provide the corresponding electropherogram (e.g., a peak graph from a Bioanalyzer) to further substantiate this result?

Response: We thank the reviewer for raising this point. We have provided the corresponding electropherogram below, which confirms the low DNA recovery after EM-seq treatment despite starting with the same amount of DNA as other methods (Response Fig. 2a-d). We were also surprised by the low DNA recovery in EM-seq and repeated the experiments multiple times, consistently observing the same result.

The reduced DNA recovery in the EM-seq method is likely due to its two-step bead purification, which may lead to greater DNA loss compared to UMBS and CBS methods. In contrast, CBS treatment requires only a single round of column desulphonation/purification. This observation aligns with the data in Figure 1d, where library yields for EM-seq are lower despite its nucleotide-preserving feature. Additionally, multiple recent studies similarly reported poor DNA recovery following EM-seq treatment,²⁻⁴ supporting our observations.

Additionally, the magnetic beads used in the EM-seq workflow may have lower purification efficiency for long, intact lambda DNA, as they were primarily developed for fragmented DNA.⁵ To investigate this, we performed a similar assay using fragmented lambda DNA treated with various methylation conversion methods. While EM-seq still showed lower DNA recovery than UMBS-seq, the recovery ratios improved compared to those observed with intact DNA, though UMBS and UBS still outperformed EM-seq (Response Fig. 2e).

Response Fig. 3: a-d, Electropherogram plots of intact DNA following treatment with the UMBS, EM, and CBS DNA methyl conversion methods, alongside an untreated control. These plots correspond to the gel images in Figure 1c. e, Comparative analysis of DNA damage in 10 ng of cfDNA induced by untreated, UMBS, CBS DNA methyl conversion methods, assessed using a TapeStation. EM-seq recovery is less than both UMBS and CBS-seq.

2. Figure 1d: While UMBS-seq generates a higher library yield with higher DNA input, its performance at lower inputs of 50 pg and 10 pg shows significant variability and, in some cases, appears to be less effective than EM-seq. Could the authors provide an explanation for this observation?

Response: We thank the reviewer for raising this point. At very low DNA inputs such as 50 pg and 10 pg, increased variability in library yield is expected due to the stochastic

nature of working with extremely limited starting material. Even minor DNA losses or fluctuations during handling and purification can lead to amplified variability after PCR amplification.

In UMBS-seq, the use of column-based purification during the desulphonation step may contribute to this variability, as column recovery efficiencies tend to fluctuate more at ultra-low DNA amounts compared to the bead-based purification employed in EM-seq. Furthermore, the higher number of PCR cycles required to generate sufficient material for sequencing from such low inputs can amplify small technical variations.

Despite these challenges at ultra-low inputs, UMBS-seq maintains competitive or superior overall performance and reproducibility at different input levels. We are actively optimizing the protocol to improve consistency for ultra-low input samples and plan to include the optimized workflow in future work.

3. The study uses lambda DNA as the standard sample to demonstrate the reduced damage and optimized conditions of UMBS-seq. However, two issues arise from this:

(1) The uniform length of lambda DNA may not fully represent the behavior of more heterogeneous samples like cfDNA. The authors are encouraged to provide a comparative analysis showing the effects of the different methods on cfDNA itself to demonstrate the preservation of DNA integrity.

Response: We thank the reviewer for this valuable suggestion. Following the reviewer's recommendation, we treated an equal amount (10 ng) of ligated cfDNA with the UMBS, EM, and UBS DNA methyl conversion methods, and assessed DNA integrity using a Bioanalyzer assay. The electropherogram plots clearly showed that both UMBS and EM treatments preserved the characteristic cfDNA size distribution and maintained good cfDNA integrity (Response Fig. 3a-c). In contrast, UBS did not retain the typical cfDNA profile, indicating greater DNA degradation and reduced integrity preservation compared with UMBS and EM-seq.

In addition, we compared the cfDNA insert size profiles of libraries prepared using UMBS-seq, EM-seq, and UBS-seq in the manuscript (shown as Figure 2d). UMBS-seq libraries exhibited an insert size distribution closely matching that of EM-seq libraries, and both were significantly longer than those generated with CBS-seq library. Consistently, we observed that UMBS-seq and EM-seq libraries yielded significantly more DNA than UBS-seq libraries (shown as Figure 2b).

In summary, these results demonstrate that UMBS-seq not only preserves DNA integrity in lambda DNA but also maintains cfDNA integrity at a level comparable to EM-seq, while outperforming UBS-seq in this regard. We have added a table as Extended Data Fig. 6 to compare different methods. We have also added to the last paragraph application recommendations.

Response Fig. 4: a-c, Electropherogram plots of cfDNA following treatment with the UMBS, EM, and CBS methylation conversion methods.

(2) In a typical library preparation workflow, DNA is first fragmented and then ligated with adapters. These fragments, much like cfDNA, are already in a small size range. When these short fragments undergo bisulfite conversion, is the resulting DNA damage as severe as that observed with the initially intact, much larger lambda DNA? A clarification on this point would be helpful.

Response: We thank the reviewer for raising this point. In our initial experiments, we used intact lambda DNA to compare DNA damage across different treatments because its large and uniform size makes differences in degradation more visually apparent.

Following the reviewer's suggestion, we performed a side-by-side comparison using fragmented and adapter-ligated DNA, which more closely resembles cfDNA in size distribution. The results were consistent with our original gDNA findings (Response Fig. 4 and Response Fig. 1b). As expected, the differences in DNA damage were less pronounced in these shorter fragments than in the much larger intact lambda DNA (Response Fig 1a and Fig. 1c) due to their reduced length and fewer potential breakage sites. However, differences between methods were still evident: UMBS-seq caused the least DNA damage, EM-seq showed moderate damage, and UBS-seq caused more damage than EM-seq.

These results confirm that the relative performance trends observed with intact lambda DNA hold true for shorter, ligated DNA fragments, supporting the robustness of our conclusions.

Response Fig. 5: TapeStation gel image of fragmented and adapter-ligated lambda DNA processed using the three indicated methods.

4. I noted that Prof. He's lab published a paper on UBS-seq in 2024 (DOI: 10.1038/s41587-023-02034-w), which introduced a solution for ultra-fast bisulfite conversion. Could the authors elaborate on the similarities and differences between UBS-seq and the current UMBS-seq? A detailed comparison of their experimental protocols, primary application areas, and respective limitations would be highly valuable.

Response: We thank the reviewer for this great suggestion. We have added a summary table comparing nearly all relevant performance metrics, including robustness, input DNA range, protocol time, DNA damage, false positives, and library complexity, as Extended Data Fig. 6 in the main manuscript to further highlight the superiority of UMBS-seq over the other tested methods (Response Figure 2). This addition also addresses a similar suggestion from Reviewer #1 to summarize and highlight the differences between UMBS-seq and UBS-seq in the manuscript.

UMBS-seq and UBS-seq were developed to address different limitations of conventional BS-seq (CBS-seq). UBS-seq primarily focused on reducing false positives and shortening protocol time, thereby improving applicability in time-sensitive settings and enabling rapid 5mC profiling. Although UBS-seq achieved lower DNA damage than CBS-seq, degradation remained substantial, limiting its suitability for low-input samples. We recommend using UBS to replace the most standard bisulfite reactions.

In contrast, UMBS-seq was specifically designed to minimize DNA damage during bisulfite treatment. While it retains the robustness and conversion efficiency of UBS-seq, it sacrifices ultra-short treatment time in favor of greatly improved DNA preservation. This makes UMBS-seq particularly well-suited for clinical applications involving low-input samples, such as cfDNA and FFPE-derived DNA, where maximizing DNA recovery is critical and treatment speed is less important.

The detailed differences between UMBS-seq and UBS-seq experimental protocols and reagent composition are summarized below and are also described in the Methods section:

1. DNA protection buffer: The UMBS-seq recipe includes a DNA protection buffer composed of an organic solvent and antioxidant to prevent DNA renaturation and reduce DNA damage, whereas the UBS-seq recipe lacks these components.
2. pH titration method: UBS-seq achieves the optimal pH by mixing two commercially available ammonium bisulfite solutions (70% and 50%) with different pH values. In contrast, UMBS-seq uses 70% ammonium bisulfite titrated with a small volume of 20 M KOH. This method allows mild bisulfite conversion at much lower temperatures and offers greater flexibility for adapting reaction conditions.
3. Temperature and reaction time: UBS-seq is performed at 98 °C for 10 min, whereas UMBS-seq is conducted at 55 °C for 90 min. This “lower temperature, long reaction time” condition minimizes DNA damage while maintaining high conversion efficiency.
4. DNA denaturation process: In UBS-seq, DNA denaturation occurs automatically due to the high reaction temperature (98 °C). In UMBS-seq, DNA is pre-denatured by a 20 min incubation at 42 °C in 0.3 M KOH, a method that causes minimal damage and avoids the harsher high-temperature treatment.

	UMBS-seq	UBS-seq	EM-seq	CBS-seq
Robustness	Yes	Yes	No	Yes
Input DNA	>0.01 ng	>1 ng	>0.1 ng	>50 ng
Protocol time	2-3 h	35 minutes	7-8 h	3-4 h
DNA damage	Low	Medium	Low	High
False positives	Near zero	Near zero	High	Moderate
Library Complexity	High	Medium	High	Low

Response Figure 2: Comparison of UMBS-seq with UBS-seq, EM-seq, and CBS-seq based on various performance metrics, including robustness, input DNA range, protocol time, DNA damage, false positives, and library complexity.

References:

1. Dai Q, Ye C, Irkliyenko I, Wang Y, Sun H-L, Gao Y, Liu Y, Beadell A, Perea J, Goel A. Ultrafast bisulfite sequencing detection of 5-methylcytosine in DNA and RNA. *Nature biotechnology*. 2024;42(10):1559-70.
2. Simons RB, Karkala F, Kukk MM, Adams HH, Kayser M, Vidaki A. Comparative performance evaluation of bisulfite-and enzyme-based DNA conversion methods. *Clinical epigenetics*. 2025;17(1):56.
3. Hong SR, Shin K-J. Bisulfite-converted DNA quantity evaluation: a multiplex quantitative real-time PCR system for evaluation of bisulfite conversion. *Frontiers in genetics*. 2021;12:618955.
4. Sawyer S, Gelabert P, Yakir B, Llanos-Lizcano A, Sperduti A, Bondioli L, Cheronet O, Neugebauer-Maresch C, Teschler-Nicola M, Novak M. Improved detection of methylation in ancient DNA. *Genome biology*. 2024;25(1):261.
5. Accomando Jr WP, Michels KB. Multiplexed reduced representation bisulfite sequencing with magnetic bead fragment size selection. *DNA Methylation Protocols*: Springer; 2017. p. 137-59.